# Development of turbulent scheme in the FLEXPART-AROME v1.2.1 Lagrangian particle dispersion model

Bert Verreyken[1,2,3], Jérome Brioude[1], and Stéphanie Evan[1]

[1]Laboratoire de l'Atmosphère et des Cyclones, UMR 8105 CNRS, University of Réunion Island, Reunion Island, France
[2]Belgian Institute for Space Aeronomy, Ringlaan 3, B-1180 Brussels, Belgium
[3]Ghent University, Department of Chemistry, Krijgslaan 281 - S3, B-9000 Ghent, Belgium

**Correspondence:** B.Verreyken (bert.verreyken@aeronomie.be)

**Abstract.** The FLEXible PARTicle dispersion model FLEXPART, first released in 1998, is a Lagrangian particle dispersion model developed to simulate atmospheric transport over large and mesoscale distances. Due to FLEXPART's success and its open source nature, different limited area model versions of FLEXPART were released making it possible to run FLEXPART simulations by ingesting WRF (Weather Research Forecasting model), COSMO (Consortium for Small-scale Modeling) or
MM5 (mesoscale community model maintained by Penn State university) meteorological fields on top of the ECMWF (European Centre for Medium-Range Weather Forecasts) and GFS (Global Forecast System) meteorological fields. Here, we present a new FLEXPART limited area model that is compatible with the AROME mesoscale meteorological forecast model (the Applications of Research to Operations at Mesoscale model)[1]. FLEXPART-AROME was originally developed to study mesoscale transport around La Réunion, a small volcanic island in the South West Indian Ocean with a complex orographic structure
which is not well represented in current global operational models. We present new turbulent modes in FLEXPART-AROME. They differ from each other by: dimensionality, mixing length parameterization, turbulent transport constraint interpretation and a novel time-step configuration. Performances of new turbulent modes are compared to the ones in FLEXPART-WRF by testing the conservation of well-mixedness by turbulence, the dispersion of a point release at the surface and the marine boundary layer evolution around Reunion island. The novel time-step configuration proved necessary to conserve the well-mixedness
in the new turbulent modes. An adaptive vertical turbulence time step was implemented, allowing the model to adapt on a finer time scale when significant changes in the local turbulent state of the atmosphere occur.

## 1  Introduction

Atmospheric transport models are divided into Eulerian and Lagrangian transport models. Eulerian models represent the at-
20 mosphere in a grid with mass being exchanged between grid cells. They are especially useful to model chemical interactions

---

[1]Applications de la Recherche à l'Opérationnel à Méso-Echelle

in the atmosphere. However, Eulerian models have difficulties maintaining the shape of narrow plumes due to numerical diffusion in their advection scheme. A number of techniques can be applied to dampen these diffusions but they generally come with great computational costs (Alam and Lin, 2008). The Lagrangian models on the other hand describe the evolution of air masses in pregenerated 3D meteorological fields obtained from a numerical weather prediction (NWP) model, allowing precise and fast modelling of atmospheric tracers released from point-sources. Uncertainties in Lagrangian models originate from linear temporal and spatial interpolation from the 3D meteorological fields of the NWP model (Stohl et al., 1995). Lagrangian particle dispersion models (LPDM) such as FLEXPART represent an air mass by a large amount of infinitesimally small air parcels, also called particles. Each individual particle is advected along the resolved wind fields with a turbulent diffusion superimposed. (Zannetti, 1990)

LPDMs are used in a variety of atmospheric studies such as source apportionment of chemical compounds (Gentner et al., 2014; Warneke et al.), studying atmospheric water vapor transport (Bertò et al., 2004; D'Aulerio et al., 2005; James et al., 2008), characterising deep stratospheric intrusions (Brioude et al., 2007; Akritidis et al., 2012), as well as hazard preparedness exercises (Stohl, 2013). Regional inverse modelling studies are also an increasingly important field of applications of LPDMs (Lin et al., 2003; Manning et al., 2003; Stohl et al., 2009; Brioude et al., 2011).

Pisso et al. (2019) describe the FLEXPART offline transport model, including the available limited area model versions. The limited area versions of FLEXPART (FLEXPART-WRF (Brioude et al., 2013), FLEXPART-COSMO (Henne et al., 2016), FLEXPART-MM5) allow particle transport in higher resolved grids to better represent mesoscale phenomena.

The AROME mesoscale forecast model has been the operation weather forecasting model at Météo France since 2008. It is designed for fine-scale modelling with grid sizes ranging from 0.5 to 2.5 km. AROME is developed by combining efforts of the French Meso-NH research model community and the ALADIN consortium[2]. Since 2015, mainland France is covered by a 1.3 km horizontally resolved grid in a Lambert conformal projection which results not only in a more realistic representation of topologically induced physical phenomena but also allows for a fine scale variation in surface types impacting for instance the sensible heat flux at the surface (MétéoFrance). FLEXPART-AROME was developed by the LACy laboratory to model particle transport around La Réunion, a french overseas territory which is covered by an AROME grid in the South-West Indian Ocean (SWIO) with 2.5x2.5 km[2] horizontal resolution in a Lambert Conformal projection. With its 90 vertical hybrid sigma levels it reaches an atmospheric altitude of about 24 km above sea level. A provisional version of FLEXPART-AROME was successfully used in the 2015 STRAP campaign to forecast transport of a volcanic plume on the Island (Tulet et al., 2017).

FLEXPART-AROME is based on the FLEXPART-WRF v3.1.3 code which is able to use the Lambert Conformal projections in the horizontal coordinate. The hybrid sigma levels are projected on Cartesian terrain-following vertical levels used by FLEXPART. To simulate turbulence induced by the complex orographic structure of the volcanic island of La Réunion and by shallow convection, we built on the turbulent modes implemented in FLEXPART-WRF by ingesting the 3D turbulent kinetic energy (TKE) field from the NWP in FLEXPART in order to harmonise turbulent motions between both.

---

[2]The ALADIN consortium contains the Algerian, Austrian, Belgian, Bulgarian, Croatian, Czech Republic, French, Hungarian, Moroccan, Polish, Portuguese, Romanian, Slovakian, Slovenian, Tunisian and Turkish weather services.

## 2 Turbulent inconsistency between NWP and LPDM

Incoherent turbulent representations may introduce unrealistic tracer transport features. For instance, if the planetary boundary layer (PBL) height is overestimated in the transport model, tracers will be advected along stronger free tropospheric (FT) winds with a different direction. If the reverse is true and the PBL height is underestimated a passive tracer released at the surface will be well-mixed over a smaller vertical range, overestimating tracer concentrations in the boundary layer.

The FLEXPART Lagrangian particle dispersion model uses the turbulent parameterization proposed by Hanna (1982) and computes the PBL top along the method of Vogelezang and Holtslag (1996). In the large-scale global grids, deep convection is a relevant sub-grid scale process. To describe this, Forster et al. (2007) adapted the convective parameterization by Emanuel and Živković Rothman (1999) in FLEXPART. Deep convection is assumed to be resolved in the mesoscale grids from AROME. The scheme was switched off by setting the LCONVECTION input parameter, introduced in FLEXPART-WRF, to zero. FLEXPART-WRF introduced two new turbulent modes using the 3D TKE fields from the NWP model. They were, however, reported to violate the well-mixedness condition, described by Thomson (1987), which states that turbulence cannot change an initially well-mixed atmospheric tracer. To resolve this in the newly implemented turbulent modes in FLEXPART-AROME, we applied the method proposed by Thomson et al. (1997), successfully used in the Stochastic Time-Inverted Lagrangian Transport (STILT) model (Lin et al., 2003), to constrain particle transport at discrete interfaces in the model.

In contrast tot he Hanna turbulence in FLEXPART, AROME TKE fields include shallow convective transport, allowing novel turbulent modes in FLEXPART-AROME to mix boundary layer air with free tropospheric air masses.

Figure 1 illustrates the difference between the TKE fields from AROME and the calculated boundary layer top[3] from FLEX-PART. We note that there is a large difference in turbulent motions in FLEXPART-WRF modes, where turbulence is only treated within the PBL, and the turbulent kinetic energy fields retrieved from AROME. The inclusion of shallow convection and convective clouds in the TKE fields will allow particles at the surface to mix to higher altitudes in the atmosphere.

## 3 Turbulent scheme development

Turbulence in FLEXPART and FLEXPART-AROME is assumed Gaussian and parametrised using a Markov process to solve the Langevin eqation. For an implementation with a discrete time step, dt, this results in:

$$\left(\frac{w}{\sigma_w}\right)_{k+1} = r_w \left(\frac{w}{\sigma_w}\right)_k + \sqrt{1+r_w^2}\zeta + \frac{\partial \sigma_w}{\partial z}\tau_{L_w}(1-r_w) + \frac{\sigma_w}{\rho}\frac{\partial \rho}{\partial z}\tau_{L_w}(1-r_w), \tag{1}$$

where $w$ is the vertical wind component of the turbulent motion, $L_w$ the turbulent mixing length, $\tau_{L_w}$ the Lagrangian time scale for the vertical autocorrelation, $\sigma_w$ the vertical turbulent velocity distribution width, $\rho$ the air density, $z$ the altitude, $r_w = \exp\left(-dt/\tau_{L_w}\right)$ the autocorrelation of the vertical wind and $\zeta$ a normally distributed random number with mean zero and unit standard deviation. The subscript $k$ and $k+1$ refer to subsequent times separated by $dt$. The first two terms on the

---

[3]Subgrid-scale orography variations and enveloping PBL height considerations, that can be taken into account in FLEXPART, are not taken into account since they don't make sense at the current mesoscale resolutions.

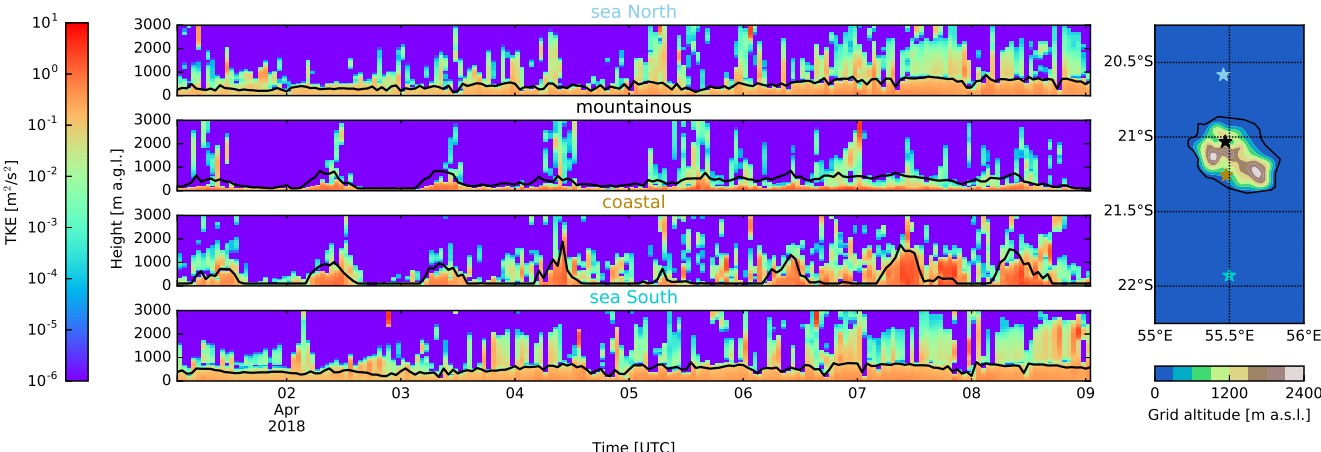

**Figure 1.** Temporal evolution of TKE fields retrieved from AROME in four different types of area around Réunion Island overlayed with a black curve showing the PBL top as calculated in FLEXPART. The vertical evolution plots on the left correspond from top to bottom with the locations indicated on the map from North to South respectively. Height of the vertical profiles is expressed in meter above ground level, over the mountainous and coastal areas this corresponds with and added 1.2 and 0.4 km above sea level respectively.

right hand side represent the native autocorrelated turbulent velocity behaviour. The third and fourth terms represent drift and density corrections respectively.(Stohl et al., 2005)

To determine $\tau_{L_w}$ and $\sigma_w$, FLEXPART-WRF has four modes defined by the TURB_OPTION input parameter introduced by Brioude et al. (2013):

5      – TURB_OPTION = 0: Turbulent velocities are set to zero.

     – TURB_OPTION = 1: Turbulence is computed using the standard FLEXPART configuration using the parameterization proposed by Hanna (1982).

     – TURB_OPTION = 2: A hybrid configuration combining TKE fields from WRF and the FLEXPART parameterization. Surface-layer scaling and local stability with the Hanna scheme determine the 3D partitioning of the turbulent kinetic 10      energy.

     – TURB_OPTION = 3: Turbulent motions are characterised directly by the TKE field from WRF and 3D partitioning is based on balancing production and dissipation of turbulent energy.

Brioude et al. (2013) reported spurious accumulation when using modes where TKE fields from the WRF are taken to characterise the turbulence.

15      In the FLEXPART-AROME code, the drift correction is set to zero and replaced by the numerical method discussed in section 3.2. Turbulent modes are extended by 24 configurations. We separated the new options according to the characteristics of each mode, these characteristics will be discussed in greater detail below. The user has a choice in the time-loop configuration, the

computation of local TKE and parameterizations for mixing length. Turbulent motions can be restricted to the vertical axis (1D), as it is in the AROME configuration over the SWIO, or partitioned in 3D using the diagnostic equations from Cuxart et al. (2000), implemented in the Meso-NH (Lac et al., 2018) mesoscale model. The 3D modes are not explicitly evaluated here but are rather implemented to anticipate future AROME developments and use of the model in combination with Meso-NH
simulations resolved on the fine-scale.

The different novel turbulent modes together with their input parameters are summarised in table A1 (Appendix A).

## 3.1 Particle time loop

FLEXPART discriminates between the particles below, and those above the PBL top. Above the PBL, particles are advanced in one user defined model synchronisation (LSYNC) time step. In the PBL, particle positions are updated along a leap-frog
between turbulent transport and resolved wind fields. The $\Delta t$ timestep, used by the leap-frog, is determined by the atmospheric stability and the user defined input parameter CTL. Vertical turbulent transport is handled in a second IFINE time loop with a time step $dt = \frac{\Delta t}{\text{IFINE}}$, where IFINE is a third user defined input parameter.

A major difference between the FLEXPART-AROME model and other FLEXPART versions is the treatment of turbulence at the PBL top. By direct use of TKE field from the NWP model, we don't characterise the PBL height explicitly. All particles
are put through the time loops. In low turbulent regions, $\sigma_w$ is small which naturally results in longer time steps:

$$\tau_w = \frac{L_w}{\sigma_w}, \qquad \Delta t = \frac{\tau_w}{\text{CTL}}, \tag{2}$$

where $L_w$ is the turbulent mixing length.

Traditionally, $dt$ is fixed over a $\Delta t$ period. However, in the new turbulent modes from FLEXPART-AROME, TKE can change abruptly, resulting in significant differences between adjacent $dt$ time steps that are not represented. To resolve this, an
adaptive vertical turbulence time step (AVTTS) was implemented. The local time step is computed as:

$$dt' = \frac{\tau_w}{\text{CTL} \times \text{IFINE}}. \tag{3}$$

After IFINE displacements, the local $dt'$ steps are accumulated in $\Delta t = \sum_{i=1}^{IFINE} dt'_i$, which is then used as the time step to displace the particle along the resolved winds.

This new time loop configuration is significantly different to the traditional fixed vertical turbulence time step (FVTTS) con-
figuration. As will be shown in section 4.1, the FVTTS is not compatible with new turbulent modes and users of FLEXPART-AROME should always use the AVTTS configuration.

## 3.2 Thomson's approach

Thomson et al. (1997) discussed the transport of particles through discrete interfaces in a random walk dispersion model. To conserve a well-mixed profile in a turbulent system with discrete TKE steps, particle transport is constrained between different
TKE regions. By imposing a net zero mass-flux at TKE interfaces in a well-mixed system and assuming maximal mixing, particles attempting to cross an interface have a probability $\alpha$ of reflection. This probability is proportional to the ratio of

Gaussian turbulent velocity distribution widths. Lin et al. (2003) introduced a correction to this probability due to density variations. In FLEXPART-AROME, this correction was not implemented as it is taken into account when solving the Langevin equation (Stohl and Thomson, 1999).

In FLEXPART-AROME, two possible interpretations of Thomson's approach have been implemented. The first considers each displacement a small discontinuity while the second arises from the grid definition of the FLEXPART-AROME model. In the small discontinuity approximation (SDA), turbulent kinetic energy is interpolated in time and space for both the initial, and the final position of a time step $dt$. The particle is supposed to cross an imaginary interface located at the middle of its trajectory. The probability of crossing is given by $\alpha = \frac{\sigma_f}{\sigma_i}$, where $\sigma_i$ and $\sigma_f$ represent the widths of the turbulent velocity distributions at the initial and final position respectively. Alternatively, one can consider the FLEXPART grid as a stack of homogeneously turbulent cells. The cell-boundaries are discrete TKE interfaces and particles attempting to cross into an neighbouring cell are reflected with a probability $\alpha$. In this mode (Step TKE), particles moving a distance $dz$ are checked to see if they cross the cell boundary. If so, the time step is split up in the time it takes for the particle to get to the boundary ($dt_1$), and the remaining time ($dt_2 = dt - dt_1$). When a particle crosses the boundary, the turbulent velocity is recalculated at the boundary to be consistent with the new local turbulence. The difference between both interpretations is visualised in figure 2.

Both options have their merit. The SDA is recommended when users are interested in a more detailed vertical profile for the FLEXPART-AROME output. Once the SDA mode is selected, users should pay attention to the IFINE and CTL parameters. If their values are low[4], the small discontinuity hypothesis no longer stands. When users want to speed up their model run and are not interested in detailed vertical distributions near the surface we suggest the use of the Step TKE option.

### 3.3 Turbulent mixing length

There are currently three parameterizations for the turbulent mixing length available in FLEXPART-AROME. The first is based on the grid size (DELTA). It is commonly used as the characteristic length scale of sub-grid eddies and is justified when the grid size falls into the inertial subrange of the turbulent flow and is recommended when the NWP model has high resolution and a nearly isotropic grid (Cuxart et al., 2000). The second parameterization is the Bougeaul-Lacarrère mixing length (BL89), a non-local turbulent mixing length parameterization proposed by Bougeault and Lacarrère (1989) that balances the TKE with buoyancy effects to determine the mixing length. This parameterization is the default mixing length used in the AROME model over the SWIO domain. The last parameterization (DEARDORFF) is the analytical limit of BL89 in a stably stratified atmospheric limit which corresponds with the results of Deardorff (1980). It was implemented to study the model behaviour in numerical tests. The use of this last parameterization is discouraged for realistic atmospheric transport. The implementation of these parameterizations is discussed in appendix B. Users of FLEXPART-AROME are encouraged to use the same mixing length parameterization as their AROME domain to get consistent results between the NWP and the LPDM.

---

[4]In our experience, we found that values of IFINE and CTL of 5 were advisable from the different tests. Simulations with CTL values of 2 showed accumulation in all modes, even when combined with IFINE values of up to 10. When using the Step TKE mode modes, we suggest not going to values of IFINE and CTL below 5 and 3 respectively. Our recommendations for the SDA mode are to keep to a minimum of 5 for both parameters.

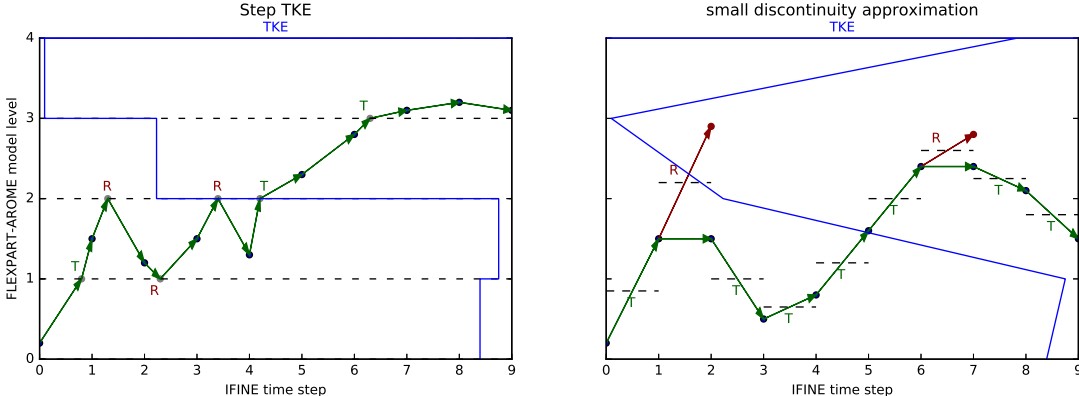

**Figure 2.** Illustrative difference between Step TKE and SDA configurations. Dashed lines represent TKE interfaces, in the Step TKE configuration they are fixed with homogeneous TKE regions inbetween, SDA interpolates TKE to the particle position and initialises an imaginary temporary TKE interface halfway the particles trajectory each step. Every time the particle tries to cross an interface we evaluate the probability of crossing and the particle will be either transmitted (T)through, or reflected (R) at the interface. The Step TKE configuration updates particle positions to the boundary before computing the probability of crossing (grey points), when particles are transmitted, their turbulent velocity is adapted to the new model layer. The SDA configuration uses a virtual position which becomes the new position upon transmission or which is never realized upon reflection (red points).

## 4   Validation

Validation tests were run using LSYNC=300, CTL=5 and IFINE=5 with output each 30 minutes during a period of 24 hours. For each test 250000 particles are initialised. The particles are not advected along resolved winds to isolate vertical turbulent motions. The horizontal domain is constrained to one AROME gridcell area over land or over sea. The output kernel of FLEX-

PART, spreading a fraction of particle mass over adjacent horizontal cells, was compensated by adding the output between adjacent cells of FLEXPART-AROME output. The grid cells over land and sea were randomly selected to perform our tests. The cell over land has coordinates 21.124S 55.379E, corresponding to a forest area on Reunion island. The cell over sea is located at 22.409S 53.939E, a cell 200 km South-West of the island. The vertical output grid goes up to 5km and is resolved by 100 m thick layers. Real TKE fields were used for the test which is why two types of area were explicitly tested. Simulations

above sea are shown here, results over land were similar unless explicitly stated otherwise. The TKE profile and the diagnosed PBL height from FLEXPART in the cell above sea are shown in Figure 3

### 4.1   Turbulent conservation of a well-mixed passive tracer

Initially well-mixed passive tracers in position and velocity space should remain unchanged in a turbulent flow. Isolating the vertical turbulence and using the MDOMAINFILL option to initialise a well-mixed passive tracer, all turbulent modes in

FLEXPART-AROME were tested. Accumulation is normalised to the initial mean mixing ratio. By using the MDOMAINFILL

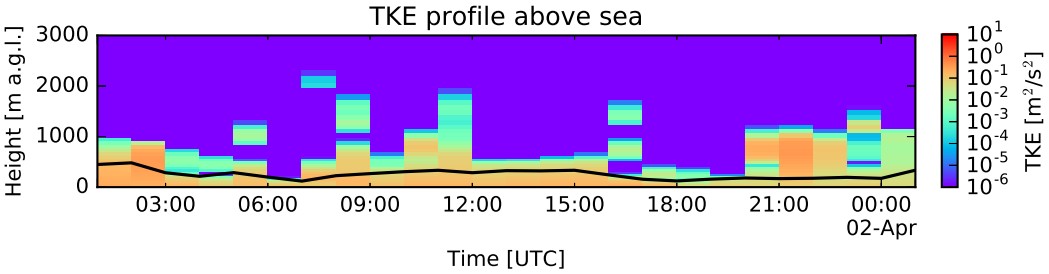

**Figure 3.** The real TKE pofile used in the validation tests above sea. The black curve corresponds to the PBL height computed by FLEXPART.

option, numerical fluctuations lead to background accumulations and dilutions of 3.5% and 4.0% respectively. Results above the sea are shown in Figure 4.

The Hanna parameterization shows systematic accumulation at the surface(11.0%). Turbulent modes introduced in FLEXPART-WRF based on TKE violate consistently the well-mixed criterion. Dilution at the surface in the hybrid FLEXPART-WRF mode is 46.4%, accumulation at the PBL top 42.3%. The results in the second FLEXPART-WRF mode are slightly better with a maximum dilution of 43.3% near the surface and an accumulation of 31.5% at the PBL top.

The AVTTS configurations perform consistently better than their FVTTS counterparts. The FVTTS result with DELTA mixing length has the largest surface accumulation of novel FLEXPART-AROME modes (surface accumulation up to 25.7%). The AVTTS DEARDORFF mode in a step TKE configuration has the least accumulation and dilution of all models (4.3% and 7.4% respectively), however, use of DEARDORFF is not recommended since it is only valid in a stably stratified atmosphere. Aside from the DEARDORFF configuration, modes combining AVTTS with BL89 best conserve the well-mixed state of the passive tracer. The step TKE option performs slightly better than the SDA in this example (0.9% less dilution and 2% more accumulation). Tests over land however showed that SDA had better results. (Appendix C)

The remaining accumulation is due to gradients in mixing length. The DELTA mode has smaller $L_w$ near the surface while DEARDORFF has larger mixing lengths at the surface compared to higher altitudes. We see that mass accumulates in these small mixing length regions.

## 4.2 Vertical dispersion of a passive surface tracer in the planetary boundary layer

The vertical dispersion of a passive surface tracer is an important test to assure efficient vertical turbulent mixing. The conservation of well-mixedness might be due to inefficient mixing and so, the surface tracer is a necessary supplementary test. We expect the tracer to be well-mixed throughout the turbulent regions within three hours after the initial release.

A point release at the surface at t=0, corresponding to 4 a.m. local time, in a FLEXPART-AROME simulation with isolated vertical turbulent motions driven with transient meteorological field was performed. The dispersion of the tracer for different turbulence modes is shown in Figure 5. The final mixing ratio profiles of are shown in appendix D.

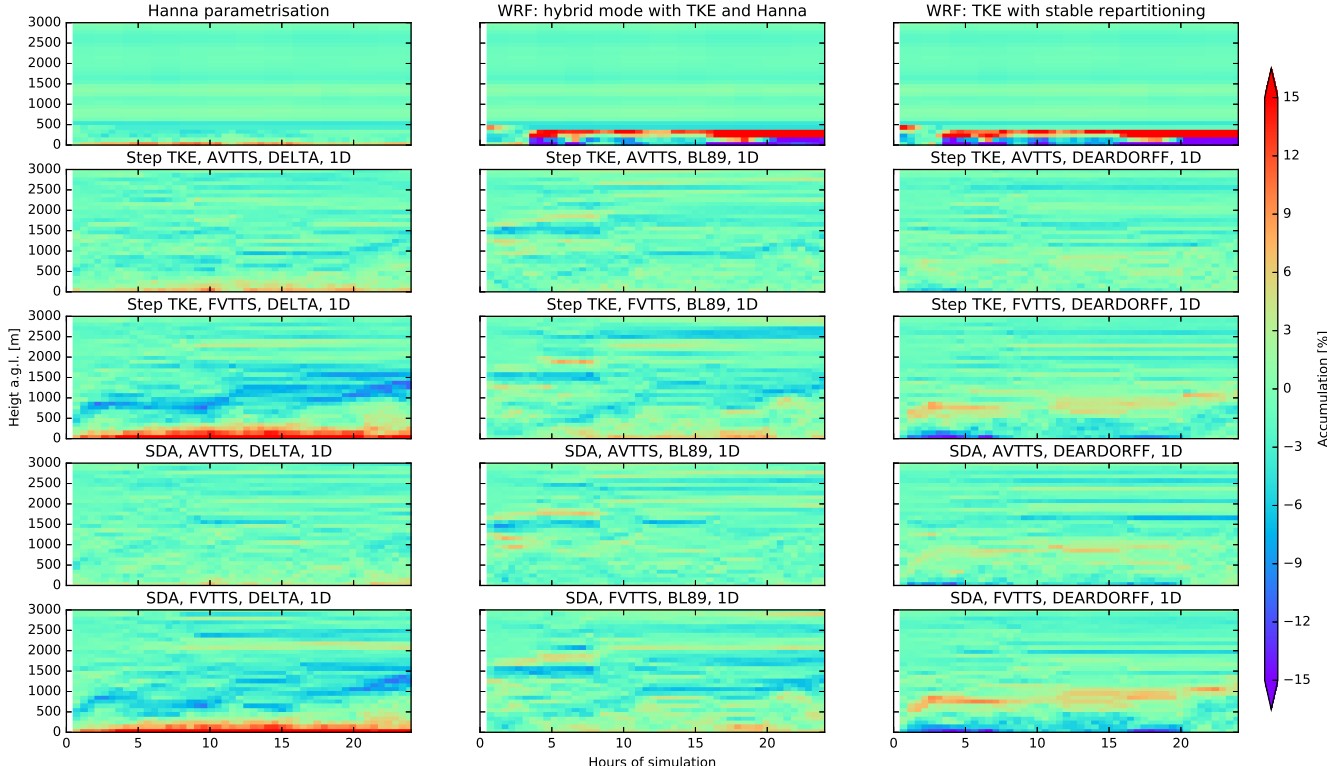

**Figure 4.** The vertical profile of accumulation in well-mixed test from all the different turbulence configurations in FLEXPART-AROME is shown throughout the 24 hour simulation test. These tests were run in a single column over the ocean surface.

In the Hanna mode and the FLEXPART-WRF modes, the tracer is mixed up to 500m above ground level within the first 3 hours. This corresponds to the maximum boundary layer top within this period. It is obvious however that the tracer is not well mixed in the FLEXPART-WRF configurations based on the turbulent kinetic energy.

Similar to the traditional configurations, the novel FLEXPART-AROME turbulent modes succeed in well-mixing the surface tracer within the first three hours. But rather than mixing up to the 500 m above ground level, where the boundary layer top is situated, the novel modes mix the tracer up to an altitude of 1000 m above ground level. This corresponds to the maximum height of the turbulent layer according to the TKE fields in the same period. There is also limited mixing between turbulent and non-turbulent regions above the shallow convective zone present in the new modes. This in contrast to the sharp PBL in FLEXPART-WRF where all particles are reflected at the PBL top in the isolated turbulence configuration. Note that the use of dynamic TKE fields result in the shifting in time of the convective zone. Particles can be mixed higher up at certain times after which they will no longer mix down but rather remain at the same position.

Due to the inclusion of shallow convective mixing in new turbulent modes, particles are allowed to breach the PBL top and near-surface concentrations in the traditional turbulent option is approximately three times larger compared to the new modes.

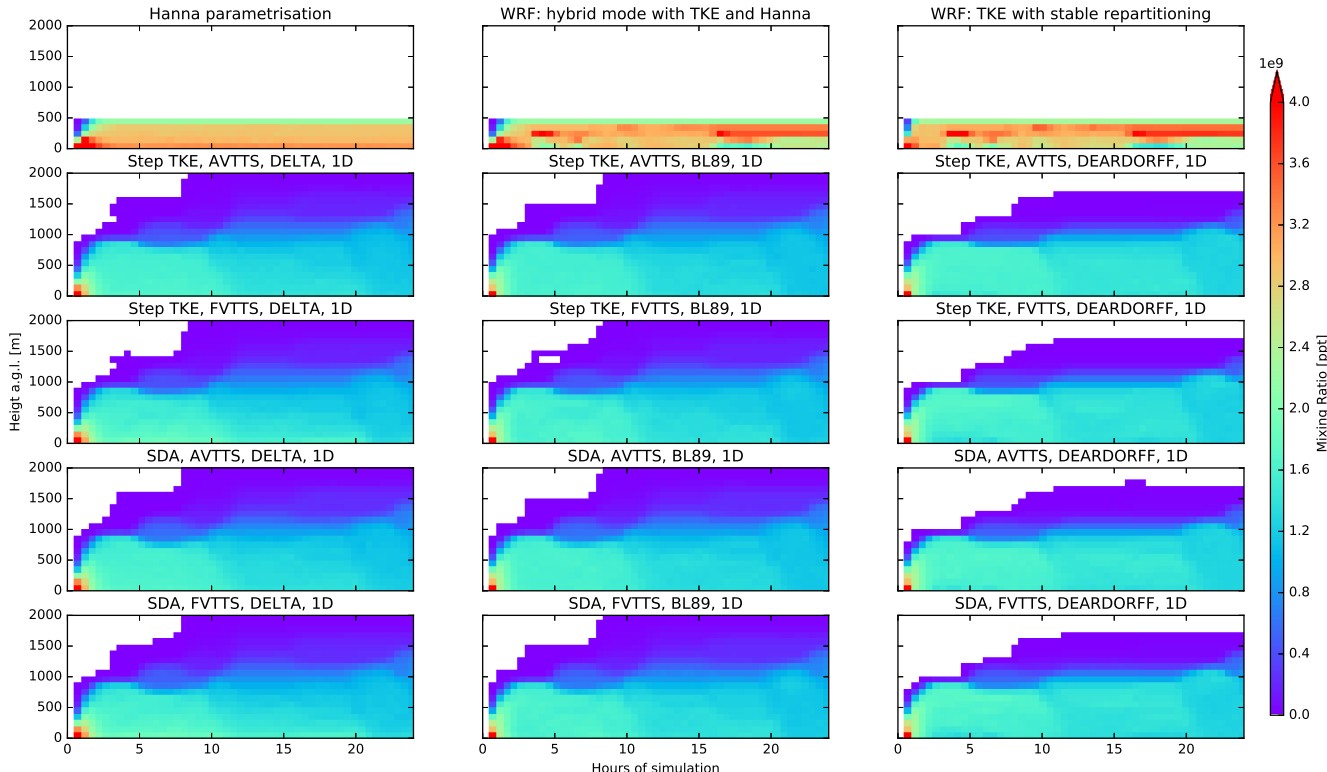

**Figure 5.** Vertical dispersion of point release at the surface are shown by the time evolution of the vertical mixing ratio profiles throughout the 24 hour simulation test for the different turbulent modes in FLEXPART-AROME. These tests were performed in a single column over the ocean surface.

The tracer is mixed over a larger vertical range causing a dilution not present in Hanna or FLEXPART-WRF turbulent modes. We highlight that, in this case, more than half of the total mass emitted at the surface is transported above the boundary layer by the new turbulent modes. This enables transport along the stronger free tropospheric winds, creating further inconsistencies in dispersion between the traditional and novel turbulent methods.

## 5 Performance

### 5.1 Marine Boundary layer tracer

FLEXPART-AROME was built to simulate atmospheric transport around Reunion Island to analyse measurements at the high altitude Maïdo observatory (Baray et al., 2013). To study the marine boundary layer (MBL) impact on measurements taken at the observatory, we continuously release a passive tracer between 0 and 5 meters above the sea with a lifetime of 24 hours. Results shown are after a spin-up time of 24 hours, LSYNC is set to 300, IFINE and CTL equal 5.

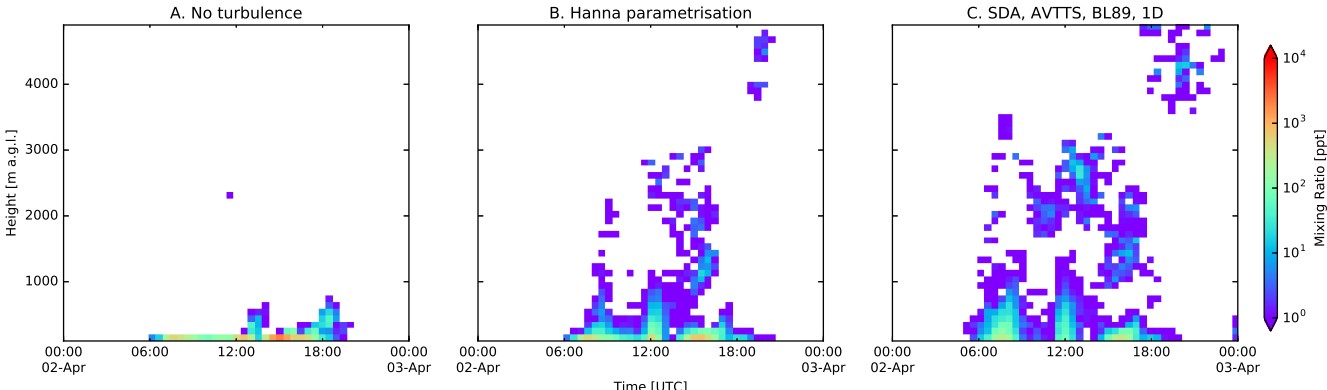

**Figure 6.** Marine boundary layer tracer profile evolution at the Maïdo observatory. On the left a simulation without turbulent motions taken into account. The middle panel shows the traditional FLEXPART turbulent mode. On the right hand side we show the results with the new turbulent mode.

Due to the strong coupling of the sea-breeze and up-slope mountainous transport the observatory is located in the MBL during the day while at night the reverse process flushes marine tracers with free-tropospheric air as found in isotopic analysis of water vapor at the Maïdo observatory by Guilpart et al. (2017). Figure 6 shows the MBL tracer at Maïdo using; i) no turbulent motions, ii) Hanna turbulence and iii) the selected new mode (TURB_OPTION=0, 1 and 111 respectively). Differences between modes with turbulence are limited in this example. The passive tracer arrives an hour earlier and has a larger vertical distribution when arriving at the observatory in the new mode compared to the performance of Hanna turbulence.

Figure 7 shows the marine boundary layer tracer above a random grid cell at sea. In this figure we clearly see the influence of clouds on the dispersion of passive marine tracer in the vertical. Tracers are convected through strong shallow convection in turbulent clouds that are not resolved in the traditional FLEXPART configuration. Surface mixing ratios in the Hanna mode are elevated compared to those obtained with the new turbulent mode as seen in the point release test.

## 5.2 Computation time

We compared the total computation time between the different simulations ran for this work. Simulations were run on a workstation with a single CPU INTEL CORE I7-7700, 32 Gb of DDR4 SDRAM with a GNU compiler. The machine was dedicated to the FLEXPART-AROME simulations to minimise the impact of parallel processes on the computation times. A complete overview of runtimes in reference to the Hanna parameterization are shown in table 1.

Traditionally particles above the PBL are not considered to be turbulent and get advected in one single LSYNC time step. In the new turbulent modes particles above the PBL top are treated in the same way as those below it. This can imply vertical turbulent loops for particles above PBL if the LSYNC input parameter is large. In the well-mixed tests we use the MDO-MAINFILL option and initialise a large amount of particles above the PBL. Due to this the relevant novel modes (excluding DEARDORFF) has a mean runtime of 4.8 times that of Hanna. We exclude DEARDORFF in this comparison since: i) its mix-

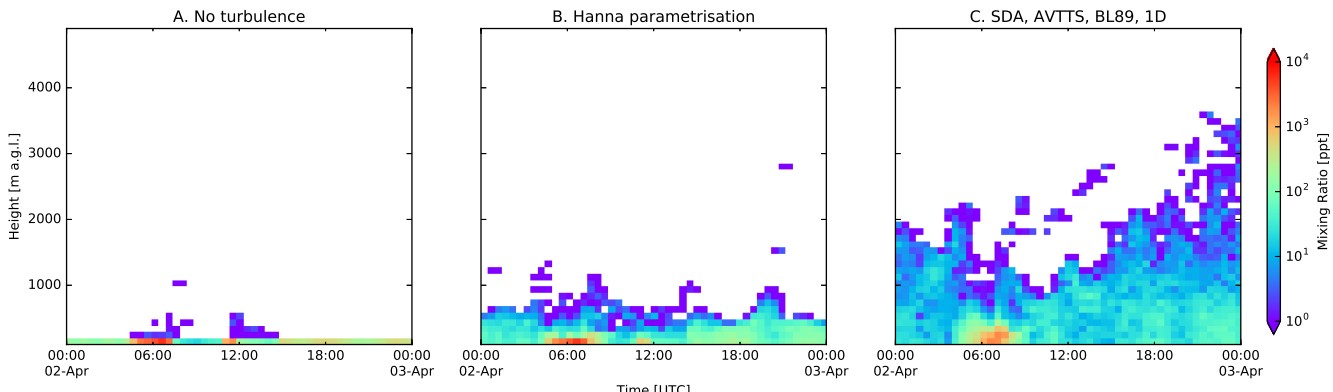

**Figure 7.** Marine boundary layer tracer profile evolution above sea. On the left a simulation without turbulent motions taken into account. The middle panel shows the traditional FLEXPART turbulent mode. On the right hand side we show the results with the new turbulent mode.

**Table 1.** Computation time ratios relative to the original Hanna parameterization computation time.

| Turbulent configuration | | | TURB_OPTION | Well-mixed test | Point release test | Marine boundary layer run |
|---|---|---|---|---|---|---|
| No turbulent motion | | | 0 | 0.96 | 2.30 | 1.89 |
| Hanna parameterization | | | 1 | 1.00 | 1.00 | 1.00 |
| WRF: hybrid mode with TKE and Hanna | | | 2 | 0.94 | 1.18 | x |
| WRF: TKE with stable repartitioning | | | 3 | 1.12 | 1.32 | x |
| Step TKE | AVTTS | DELTA | 10 | 4.95 | 1.06 | x |
| | | BL89 | 11 | 4.89 | 1.06 | x |
| | | DEARDORFF | 12 | 6.81 | 1.06 | x |
| | FVTTS | DELTA | 20 | 4.95 | 1.04 | x |
| | | BL89 | 21 | 4.99 | 1.05 | x |
| | | DEARDORFF | 22 | 6.44 | 1.02 | x |
| SDA | AVTTS | DELTA | 110 | 4.95 | 1.19 | x |
| | | BL89 | 111 | 5.21 | 1.32 | 1.37 |
| | | DEARDORFF | 112 | 9.05 | 1.24 | x |
| | FVTTS | DELTA | 120 | 5.20 | 1.17 | x |
| | | BL89 | 121 | 5.58 | 1.31 | x |
| | | DEARDORFF | 122 | 8.57 | 1.16 | x |

ing length has no lower limit except the implicit limit imposed by limiting the minimum time step and ii) it's use is discouraged since the mixing length is only valid in very specific cases. The DEARDORFF modes have a runtime of 7.5 times the Hanna runtime in testing the well-mixedness.

When running the point release the relevant new modes are 15% slower than the original mode. In the marine boundary layer, the turbulent mode combining the SDA, AVTTS, and BL89 options in a 1D configuration ran 37% longer than the Hanna parameterization. We also remark that no turbulent parameterization leads to longer run times in these two tests. This is due to the straightforward implementation of turbulent velocities being set to zero. Time steps in displacing the particle are conserved and since the vertical turbulent dispersion is not represented particles remain in regions with a very low time step.

## 6 Conclusions

We developed the new FLEXPART-AROME limited domain model version of FLEXPART based on FLEXPART-WRF v3.1.3. This configuration was build to model transport around Reunion Island in the Indian Ocean, a small volcanic island which has a complex orographic structure, but can be used with any AROME domain. To simulate turbulence consistently with the operational meteorological model in the region, we implemented new turbulent modes that ingest 3D TKE fields from the NWP. Due to shallow convection being taken into account in determining the TKE fields in AROME, FLEXPART-AROME is able to represent sub-grid scale shallow convective features. There are three important developments that users should consider when selecting the turbulent option that best suits their needs.

- To better represent the local turbulent state of a particle, an adaptive time step was implemented. This configuration is referred to as the adaptive vertical turbulence time step approach and performs consistently better in conserving the well-mixed state of the atmosphere compared to the traditional configuration.

- Turbulent drift in the model is numerically constrained by using the formalism introduced by Thomson et al. (1997). It consists in reflecting or transmitting particles at discrete turbulent interfaces to conserve the well-mixed state of an initially well-mixed atmosphere. Two possible interpretations of this formalism have been implemented. One approximates turbulence in the FLEXPART-AROME grid by considering every grid-cell to have uniform turbulence with transport being constrained at the boundaries of the model grid and is referred to as the Step TKE option. The other uses the small discontinuity approximation where the turbulent profile is vertically interpolated and transport is constrained at each displacement. The latter is referred to as the SDA option. When users are interested in vertical output grids with high resolution, as in the AROME grid, we advise to use the SDA option. If not, users can select the Step TKE option with lower values of the IFINE and CTL input parameters to speed up the model.

- Three different mixing length parameterizations are implemented: DELTA, BL89 and DEARDORFF. Use of the last parameterization is discouraged due to it only being valid in stably stratified atmospheres. Users are encouraged to adapt the choice of mixing length parameterization to be in accordance with the NWP.

New turbulent modes have a computation time that is about 5 times larger compared to the Hanna parameterization when a large fraction of the particles are above the PBL. However, simulation of tracers predominantly present in the PBL using a new mode in the AROME SWIO domain only take 15% longer than the original configuration.

FLEXPART-AROME will be used to study the arrival of marine boundary layer tracers at Maïdo observatory on Reunion
island, and the vertical distribution of marine aerosols above the ocean in comparison with measurements. Ingestion of meteorological fields coming from the Meso-NH mesoscale research model will also be introduced in the future to simulate transport at higher resolutions around La Réunion to help study air mass transport on a case study basis.

*Code availability.* The FLEXPART-AROME code is openly accesible on FLEXPART.eu

*Data availability.* Data used for the different tests is available upon request.

**Appendix A: Different turbulent modes and their respective input parameters**

Table A1 shows the different novel turbulent modes implemented in the FLEXPART-AROME code.

**Table A1.** Different turbulent options introduced in FLEXPART-AROME and their configuration.

| TURB_OPTION | | AVTTS | | FVTTS | |
| --- | --- | --- | --- | --- | --- |
| | | 1D | 3D | 1D | 3D |
| | DELTA | 10 | 15 | 20 | 25 |
| Step TKE | BL89 | 11 | 16 | 21 | 26 |
| | DEARDORFF | 12 | 17 | 22 | 27 |
| | DELTA | 110 | 115 | 120 | 125 |
| SDA | BL89 | 111 | 116 | 121 | 126 |
| | DEARDORFF | 112 | 117 | 122 | 127 |

**Appendix B: Implementation of the turbulent mixing length parameterizations**

The importance of turbulent mixing length in the new modes is the closing of the turbulent parameterization. Without this value, we have no information on how far particles can mix and so we would have no information on the turbulent time scale.
There are three different implementations of turbulent mixing length $L_w$. The 1D DELTA $L_w$ is computed as follows:

$$L_w(\text{DELTA},1\text{D}) = \min(0.4 * h(k), \Delta z(k)), \tag{B1}$$

where $h(k)$ and $\Delta z(k)$ represent the height and the thickenss of th k'th model layer respectively. When simulations are run in the 3D mode we use the following formula:

$$L_w(\text{DELTA,3D}) = \min\left(0.4 * h(k), \sqrt[3]{\Delta x \Delta y \Delta z(k)}\right),$$
(B2)

where $\Delta x$ and $\Delta y$ represent the horizontal resolutions.

The DEARDORFF parameterization is computed by:

$$L_w(\text{DEARDORFF}) = \begin{cases} \sqrt{\frac{2\text{TKE}\theta_{v,ref}}{g\partial\theta_v/\partial z}}, & \text{if} \quad \partial\theta_v/\partial z > 0, \\ \Delta z(k), & \text{otherwise.} \end{cases}$$
(B3)

Here, TKE is the local turbulent kinetic energy, $\theta_{v,ref}$ is the virtual potential temperature of the reference state, $\partial\theta_v/\partial z$ is the vertical gradient of the virtual potential temperature and g is earth's gravitational acceleration constant. In FLEXPART-AROME however, the virtual potential temperature is approximated by the potential temperature, neglecting the humidity
effect on the air masses.

The BL89 parameterization computes the distance that an air parcel can travel upward and downwards by using the local turbulent kinetic energy and combines both to compute the turbulent mixing lenght:

$$TKE = \int_z^{z+l_{up}} \frac{g}{\theta_{v,ref}}(\theta(z') - \theta(z))dz',$$
(B4)

$$TKE = \int_{z-l_{down}}^z \frac{g}{\theta_{v,ref}}(\theta(z) - \theta(z'))dz',$$
(B5)

$$L_w(\text{BL89}) = \left(\frac{l_{up}^{-2/3} + l_{down}^{-2/3}}{2}\right)^{-3/2}.$$
(B6)

These equations are solved on the discrete model layers. As a consequence, the minimal mixing length equals $\Delta z$. Similar as in the DEARDORFF parameterization, the virtual potential temperatures are approximated by the potential temperatures. The
1D and 3D parameterizations do not differ for both the DEARDORFF and the BL89 parameterizations.

It is important here to note that the DEARDORFF parameterization is the only parameterization that does not have a lower limit based on the grid definition. It only falls back on the minima of the other implementations when its value becomes negative. The lower limit is rather a computational remnant which stems from the minimal time step. In equation 3 the $dt'$ has a fixed minimum which means that the turbulent time scale is numerically forced to a specific value. When computing $\tau_w$ in
equation 2 the $\sigma_w$ value is fixed by the input which means that when its value is forced by the algorithm, we artificially adapt the turbulent mixing length.

## Appendix C:  Conservations of well-mixedness over land

Shown in Figure C1 is the conservation of well-mixedness over land in the morning when the PBL is growing. We see that the DELTA modes all have some accumulation near the surface, the AVTTS SDA mode having the least accumulation, similar to the stable PBL over sea. A surface accumulation over land in Hanna in the bottom layer of maximum 14.5%. Comparing the best performing relevant TURB_OPTION parameters 11 and 111 we see that the accumulation in the step TKE mode near the surface is 2.0% larger with the accumulation occurring at the surface from 10 hours simulation onward.

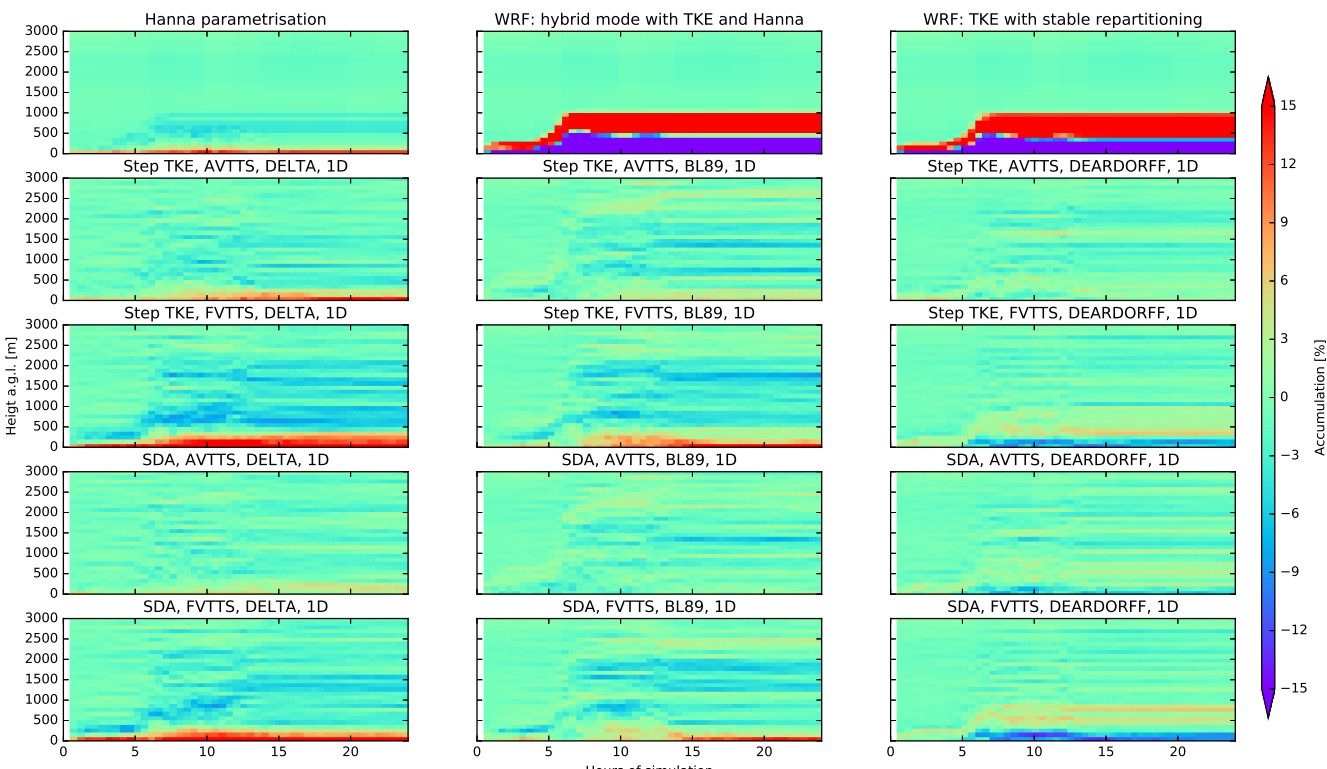

**Figure C1.** Accumulation in well-mixed test in all different turbulence configurations in FLEXPART-AROME. These tests were run in a column over the ocean surface.

## Appendix D:  Conservations of well-mixedness over land

After the 24 hour simulation of a passive tracer released at the surface, final mixing ratio profiles for all tested turbulent modes are shown in Figure D1. Due to the shallow PBL in the traditional modes the mixing ratios of the FLEXPART-WRF configurations are a factor 2 to 3 larger. The new turbulent modes are all well mixed near the surface. Due to the shifting convective zone near the top there is no sharp difference between PBL and FT.

We can clearly see two different kinds of mixing between the DEARDORFF parameterizations on the one hand and the DELTA and BL89 modes on the other hand. While DEARDORFF is based on an analytical formula with no real lower limit except the one implicitly imposed by the minimal time step, vertical mixing above the more turbulent layer is slower. This results in a mixing ratio profiles which do not reach as high as the other modes who's lower limit on turbulent mixing length is based on the grid definition.

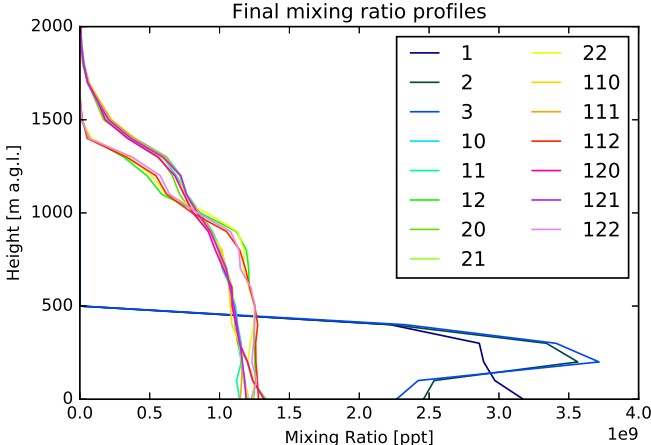

**Figure D1.** Final mixing ratio profiles in the surface tracers test released over sea. The legend shows the numerical value of the TURB_OPTION parameter input.

*Author contributions.* Jérome Brioude developped the provisionary FLEXPART-AROME version and adapted FLEXPART-WRF code to ingest AROME data. He supervised and advised Bert Verreyken who was responsible for implementing and testing the Thomson methodology to use 3D TKE fields in the model. Stéphanie Evan was developer on the FLEXPART-WRF version used as a base and a sought after consultant on development of FLEXPART-AROME.

*Competing interests.* The authors declare that they have no conflict of interest.

*Disclaimer.* FLEXPART-AROME is distributed in the hope that it will be useful, but WITHOUT ANY WARRANTY; without even the implied warranty of MERCHANTABILITY or FITNESS FOR A PARTICULAR PURPOSE. See the GNU General Public License for more details.

*Acknowledgements.* This study has been supported by the project OCTAVE of the "Belgian Research Action through Interdisciplinary Networks" (BRAIN-be) research programme (2017-2021) through the Belgian Science Policy Office (BELSPO) under the contract number BR/175/A2/OCTAVE. We thank Météo France for sharing the AROME output files from forecasts in the South-West Indian Ocean which made this work possible.

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
