# Peer review of "Development of turbulent scheme in the FLEXPART-AROME"

_Geoscientific Model Development, 2019_

## Referee Comment (RC1) · Anonymous Referee #1 · 8 May 2019

This manuscript presents a new development of the FLEXPART model to use the mesoscale AROME model. The main interest is the detailed attention paid to turbulent mixing and the development of schemes to correct problems that many users may not even be aware of. The work is thorough and useful, I am happy to recommend publication.

The main comment I would have concerns the treatment of mixing in convective clouds. Most of the paper appears to be written without this in mind, but some sections seem to suggest it is a dominant factor. Maybe this can be clarified in the text with an expanded discussion and some comparisons. Given the importance of clouds, should model

comparisons not be shown with and without clouds?

Minor comments:

The captions of Figures 4 and 5 could be expanded. The explanation of the figures in the text could also be expanded for clarity.

I was not sure what the purpose of giving the input codes in Table 1 is. Maybe some useful information can be given to help the reader keep in mind the difference between the schemes?

Page 7, Line 5: The explanation of bottom-up and top-down could be made clearer. I even wonder about the terminology – maybe a better name could be found for these, and for the turbulence scheme. Section 3.3: As for the comment regarding bottom-up/top-down, the turbulence schemes could be better explained.

Writing comments:

The paper needs a more thorough round of proofreading, eg.:

"Provisional" not "provisionary"

"LPDM" not "LDPM" (1 instance).

Page 3, Line 6: The phrasing is odd: WRF-Flexpart did not exist at the time of the Thompson publications.

Page 3, Line 9: sentence fragment.

Capitalizations: "TKE" not "tke". Please be consistent when referring to figures, either: "Figure 3" or "Fig. 3", preferably capitalized.

"iterpretations"

---

## Referee Comment (RC2) · Anonymous Referee #2 · 24 Jun 2019

The manuscript by Verreyken and co-workers presents the development of a novel version of the widely-used Lagrangian particle dispersion model FLEXPART for the use with output from the limited area model AROME. However, the focus of the manuscript is on the development and validation of new turbulence schemes that could be used as an alternative to the default FLEXPART parameterisation that is not building on the turbulence information available from the driving meteorological model. The design and implementation of new turbulence schemes for FLEXPART certainly is an important development and may lead to improved dispersion simulations in many different areas of atmospheric transport. However, I feel that the study was not carried out with the required care and thoroughness and needs major revisions before it can be accepted

in GMD. A detailed list of my concerns follows.

Major comments

Validation

A number of different new turbulence schemes (or various settings for these) were tested for conservation of well-mixedness and for two different case studies with surface releases. Although, the well-mixedness test is very important, it alone does not seem to be sufficient to judge which setup performance best. The surface release cases remain very qualitative in their evaluation. I had hoped for a more quantitative validation of the turbulence schemes, either by application to existing tracer release experiments or, if this is not easily possible, by an application to real-world observations made at the Maido observatory on La Reunion. The manuscript mentions observations of water vapor isotopes at the observatory, but surely there are also other observed tracers that could be used to identify the PBL influence at the site and could be compared to different transport simulations in a more quantitative way and under different mixing conditions (if possible). Such an analysis is hinted to at the end of the conclusion as part of future work, but I strongly feel that the current manuscript requires this more quantitative validation as well. Without such an analysis I don't think a clear conclusion can be drawn in terms of which turbulence scheme should be used in future applications and if the new schemes are even performing correctly at all.

Structure and Presentation

It is not always easy to follow the flow of the manuscript. It is often not well explained why and how certain things were done (see examples below). In other sections the manuscript is lacking the degree of detail that is important for a model development paper. Also, the current conclusions are lacking a clear recommendation, which of the new turbulence schemes should be used in future studies, and if the developments presented here will make it back into the main and/or WRF FLEXPART versions.

Minor comments

Abstract, last sentence: This was said before. Maybe reformulate to make it the main conclusion of the study. Also include a statement/recommendation of the default turbulence scheme to be used with FLEXPART-AROME.

Introduction: The pros and cons of Lagrangian versus Eulerian models are stated. However, this section is lacking good citations and it is also a bit too negative about Eulerian models. For example advection schemes in Eulerian models can be designed in ways that they are conserving mass and are less diffusive. But usually this comes with a prize of higher complexity and larger computational costs. But generally the statement that Eulerian models cannot do this is not valid. Also, there are other uncertainties connected to offline Lagrangian transport models. 1) temporal resolution of input meteorology, when running off-line, 2) less explicit description of turbulence (in comparison to prognostic TKE in most NWPs), exactly what the manuscript highlights later.

P2,L6ff: Regional inverse modelling studies are also an increasingly important field of application of LPDMs: eg. Stohl et al. (2009), Lin et al. (2003), Manning et al. (2003)

P2,L11f: Does this sentence still refer to different FLEXPART versions? Please mention these again with reference. Next to FLEXPART-WRF, there is also the FLEXPART-COSMO version mentioned in Pisso et al. (2019) and described in Henne et al. (2016).

P2,L15: "French metropolitan area" Does this refer to Paris or the whole mainland France domain?

P2,L20: Is AROME-SWIO also an operational model product by MeteoFrance?

P2,L24: Reference to FLEXPART-WRF publication missing. On which FLEXPART-WRF version is FLEXPART-AROME based?

P3,L9f: The sentence is incomplete. I guess it should continue after (fig 1) without starting a new sentence.

[Figure]

PBL diagnostics: Why is the PBL height diagnostic, which was solely based on theta_v, called robust as compared to the Richardson bulk number approach in FLEXPART? The latter is also using the theta_v profiles from AROME, but in addition it also uses wind shear information (again from AROME). Compare Stohl et al. (2005). However, FLEXPART in its original version uses an "enveloping" PBL height, which is the maximum from the neighboring grid cells and the two model time steps in memory. It also extends the PBL height in areas with large subgrid-scale orographic variability. Both approaches may NOT be justifiable for high resolution simulations and may be the reason for the "overestimation" of PBL heights by FLEXPART in mountainous terrain (P3,L11). Which approach was followed in FLEXPART-AROME? The same as in FLEXPART-ECMWF? It is also not clear how PBL heights were estimated solely based on theta_v. By a parcel method? Assuming that the PBL height is the height where theta_v is first larger than theta_v at the surface including a surplus surface temperature? Was an additional interpolation between model levels used (like in FLEXPART)?

Comparing turbulent layers and PBL heights: The TKE layer diagnosed from AROME is strictly speaking not the same as the classical PBL height. Hence, I suggest to clearly separate the naming from what is otherwise called PBL height. This is implicitly introduced in the description, but it would be better to clearly distinguish between this turbulent layer and the PBL! It would be good to clearly define this layer in section 2 and explain in more detail how it was diagnosed from the model output. Currently this is only done in the caption to Figure 2 although the resulting layer depth is already displayed in Figure 1. From this it is also clear that one cannot conclude from the comparison between turbulent layer and PHL heights that the latter is under- or overestimated (last sentence section 2). One can only say that the former is greater or smaller than the other.

P3,L15: In FLEXPART it is also possible for particles to cross from the PBL to the FT through the subgrid-scale convection scheme. Was this switched on or off for FLEXPART-AROME. The scale would probably call for switching it off but this was not

clearly stated anywhere?

P3,L21: This "erratic behavior" could be avoided by detecting the layer top where at least two neighboring levels show TKE below the threshold value. From the examples given in Fig 2. It seems it is always a single level with low TKE between PBL and shallow convection zone. Does the erratic behavior actually matter for the TKE-based turbulence scheme in FLEXPART-AROME? Or is it only a diagnostic for the comparison with FLEXPART's PBL height estimation? This should be clarified in the text as well.

One last question concerning the TKE layer height: From the layer heights displayed in Fig. 1, my conclusion would be that mixing over the sea is more intensive or at least reaches higher than over the mountains. This is counter-intuitive, but possibly related to the fact that heights above ground are shown. What are the model orography heights for the 4 points for which the layer heights were evaluated in Fig 1?

P5,L15: At this point not clear what a Thomson interface is and there is no reference given either.

P5,L15 and section 3.1: How does the Thomson approach actually justify setting the density correction to zero? The density is not affected by turbulence intensity and as such density gradients are not explicitly treated by the Thomson approach.

P5,L18ff: Point out that this choice refers to the 1D, 3D options in Table 1. Please give a reference to the "diagnostic equations from Meso-NH" so that these can be found by the interested reader or even repeat them here if they are central.

P7,L3ff: The two different ways how to calculate the time step should be introduced much clearer and the terms 'bottom-up' and 'top-down' properly introduced as two ways how to calculate the turbulent time step. These terms have different meanings in different fields and in the context of the time step it remains a bit unclear why they were chosen.

Turbulent mixing length: Would it be possible to give the equations for the three different

ways how L_w was calculated?

Section 4, Validation: The setup should be described with more care and detail. For example: Were mean wind fields set to zero for this case? Were actual fields from AROME used for this exercise or some standard fields? Does it matter which exact locations were chosen? The location should not matter, only the surface properties, if winds were set to zero. How many particles were used in these exercises? How many per grid column? What was the FLEXPART PBL height in these grid cells and how did the TKE profile look like. Maybe both could be added to Fig 4.

Section 4: The 3D configurations of the turbulence scheme are never discussed only introduced in section 3. Were these schemes not tested after all? If not, why introduce them in the methods sections? If they were tested, how did the results compare to the 1D cases?

P8,L26: 'TURB_OPTION=11 and 111'. In the previous sentence the settings were spelled out not just the option index given. To keep the text flowing the same should be done here. The option index could still be given in braces (wherever this helps to clarify things). The sentence is also a bit odd, because already the previous sentence said 'DEARDORFF [...] has the least accumulation', which is the same as 'best conserve well-mixed state'. Better start with 'Besides the DEARDORFF modes, modes xxxx best coserve ...'.

P8,L29ff: The discussion on L_w could be more illustrative if profiles of L_w could also be included in Fig 4 or together with PBL heights and TKE profiles in a supplement.

Figure 4/5: Why is the second WRF TKE mode called 'stable repartitioning' here? It was introduced as a 'TKE only' method above. No mention of it being stable. In the caption it should be repeated that these are results for a grid cell over the ocean.

P9,L4: I think this should be 'Near-surface concentrations".

Section 4, Fig 5: How is it possible that the vertical gradients in the simulations with

new turbulence scheme are maintained over time. Even if the mixing in the shallow convection zone is smaller than in the PBL, one would expect that the vertical gradient eventually vanishes in the 24 hours of simulation, as it quickly does in the Hanna case within the PBL. In order to illustrate this more clearly, it would also be interesting to see the final mixing ratio profiles of all configurations in a comparison.

Section 5: It would be valuable if this section would show some kind of comparison with observations at Maido. Such a comparison could help supporting the authors suggestion that the new turbulence modes are superior to previous schemes. Without such a comparison there is little evidence that the performance of the new schemes is more realistic.

P10,L3: It is always confusing with LPDMs to write about 'particle transport' which can easily be confused with aerosol particle transport. Here, it would probably work best to simply replace 'particle transport' by 'atmospheric transport'.

Figures 6/7: Most of the text of the figure captions is also stated in the main text. Remove from caption. Rather repeat what is seen in each sub-panels. Sub-panels should also be labeled by letters.

Section 5.2: This section needs some introduction on how computation times were estimated. Were repeated runs carried out for each configuration? This is important since run-times may differ due to other processes running on the same machine and/or I/O may be influenced by other processes. It would also be helpful to mention on what architecture and with which compiler (options) and with which parallelisation approach these results were obtained. Do these timings reflect run times for the complete model runs or just for the transport part of the model? Please speculate why the increase in computation time was so much larger for the well-mixed test compared to the surface release? Why were computation times larger for the no-turbulence cases? Shouldn't these perform much faster, since only the mean motion needs to be solved for (which was zero in the well-mixed and point release tests)? I see it is explained later on. So

only a quick-and-dirty implementation of no turbulence was used. But then one should not compare these run-times. There is little value in it since they present some kind of artificial, never-used option.

Conclusions: These are a bit non-conclusive. So what is the recommendation for future use of the model? Which turbulence mode should be used and why?

P13,L6: Shouldn't this be 3D TKE fields? Which dimension would be dropped out for them to be 2D?

Technical comments

P2,L4: I think the sentence makes more sense if "into the atmosphere" is removed.

Figure 1: Star for mountain location almost invisible. Use different colour instead.

P3,L30: Use braces around the argument to exp.

P9,L7: 'en' should be 'and'.

P10,L4: Maido is spelled differently at different locations in the manuscript. Please unify.

References

Henne, S., Brunner, D., Oney, B., Leuenberger, M., Eugster, W., Bamberger, I., Meinhardt, F., Steinbacher, M., et al.: Validation of the Swiss methane emission inventory by atmospheric observations and inverse modelling, Atmos. Chem. Phys., 16, 3683-3710, doi: 10.5194/acp-16-3683-2016, 2016.

Lin, J. C., Gerbig, C., Wofsy, S. C., Andrews, A. E., Daube, B. C., Davis, K. J., and Grainger, C. A.: A near-field tool for simulating the upstream influence of atmospheric observations: The Stochastic Time-Inverted Lagrangian Transport (STILT) model, J. Geophys. Res., 108, 2003.

Manning, A. J., Ryall, D. B., Derwent, R. G., Simmonds, P. G., and O'Doherty, S.:

Estimating European emissions of ozone-depleting and greenhouse gases using observations and a modeling back-attribution technique, J. Geophys. Res., 108, 2003.

Stohl, A., Seibert, P., Arduini, J., Eckhardt, S., Fraser, P., Greally, B. R., Lunder, C., Maione, M., et al.: An analytical inversion method for determining regional and global emissions of greenhouse gases: Sensitivity studies and application to halocarbons, Atmos. Chem. Phys., 9, 1597-1620, doi: 10.5194/acp-9-1597-2009, 2009.
* * *

---

## Author Comment (AC1) · 9 Aug 2019

- We would like to thank the 2 reviewers for their helpful comments. The manuscript has been revised according to the referees' comments. For clarity we repeated the referee comments and respond to them below each one. Our responses are indicated by starting the paragraph with a dash. A marked-up manuscript version of the revised manuscript is included as supplementary material.

Anonymous Referee #1

This manuscript presents a new development of the FLEXPART model to use the

mesoscale AROME model. The main interest is the detailed attention paid to turbulent mixing and the development of schemes to correct problems that many users may not even be aware of. The work is thorough and useful, I am happy to recommend publication.

The main comment I would have concerns the treatment of mixing in convective clouds. Most of the paper appears to be written without this in mind, but some sections seem to suggest it is a dominant factor. Maybe this can be clarified in the text with an expanded discussion and some comparisons. Given the importance of clouds, should model comparisons not be shown with and without clouds?

- The FLEXPART model includes a deep convective scheme that redistributes the particles vertically based on the parameterisation by Emanuel and Živković-Rothman (1999). This scheme was not used in our simulations because at a resolution at 2.5x2.5km^2, one can assume that the vertical transport is resolved by AROME. The mixing in convective clouds is treated, to some extent, in FLEXPART-AROME by using the TKE fields from AROME that includes the turbulent mixing from deep and shallow convection. Although mixing in convective clouds is a major advantage of the novel turbulent scheme compared to the old, it was not the main focus of the study. Since AROME is an operational model, it is not possible for us to do an intensive comparison between simulations with and without clouds.

Minor comments:

The captions of Figures 4 and 5 could be expanded. The explanation of the figures in the text could also be expanded for clarity.

- The manuscript has been changed.

I was not sure what the purpose of giving the input codes in Table 1 is. Maybe some useful information can be given to help the reader keep in mind the difference between the schemes?
- The main purpose of Table 1 is to help future users of FLEXPART-AROME to navigate the different turbulent options. The table has been moved to the annexes since it contains no added value to the main text. Thank you for this remark.

Page 7, Line 5: The explanation of bottom-up and top-down could be made clearer. I even wonder about the terminology – maybe a better name could be found for these, and for the turbulence scheme. Section 3.3: As for the comment regarding bottomup/top-down, the turbulence schemes could be better explained.

- We have changed the terminology and altered the discussion to better express the difference between both configurations.

Writing comments:

- Thank you for your thoroughness, the corrections have been taken into account.

Anonymous Referee #2

The manuscript by Verreyken and co-workers presents the development of a novel version of the widely-used Lagrangian particle dispersion model FLEXPART for the use with output from the limited area model AROME. However, the focus of the manuscript is on the development and validation of new turbulence schemes that could be used as an alternative to the default FLEXPART parameterisation that is not building on the turbulence information available from the driving meteorological model. The design and implementation of new turbulence schemes for FLEXPART certainly is an important development and may lead to improved dispersion simulations in many different areas of atmospheric transport. However, I feel that the study was not carried out with the required care and thoroughness and needs major revisions before it can be accepted in GMD. A detailed list of my concerns follows.

Major comments:

Validation: A number of different new turbulence schemes (or various settings for these) were tested for conservation of well-mixedness and for two different case studies with surface releases. Although, the well-mixedness test is very important, it alone does not seem to be sufficient to judge which setup performance best. The surface release cases remain very qualitative in their evaluation. I had hoped for a more quantitative validation of the turbulence schemes, either by application to existing tracer release experiments or, if this is not easily possible, by an application to real-world observations made at the Maido observatory on La Reunion. The manuscript mentions observations of water vapor isotopes at the observatory, but surely there are also other observed tracers that could be used to identify the PBL influence at the site and could be compared to different transport simulations in a more quantitative way and under different mixing conditions (if possible). Such an analysis is hinted to at the end of the conclusion as part of future work, but I strongly feel that the current manuscript requires this more quantitative validation as well. Without such an analysis I don't think a clear conclusion can be drawn in terms of which turbulence scheme should be used in future applications and if the new schemes are even performing correctly at all.

- The principle of the turbulent scheme developments for FLEXPART-AROME was to improve consistency between the NWP and the offline LPDM dynamics. The focus of our development efforts was thus numerical consistency when looking at the well-mixedness criterion and the surface release test. We assume that any further testing would not probe the FLEXPART-AROME developments but rather the differences between the online and offline turbulent parametrizations of AROME and FLEXPART respectively. Furthermore, existing tracer release experiments were not conducted within any available AROME domains. Characterising the influence of the PBL development at Maïdo observatory on observations is an ongoing research project which will also rely in part on FLEXPART-AROME and the Meso-NH mesoscale simulations during the intensive observation period of the OCTAVE project (March to May 2018). We think that an analysis of the influence of PBL development on Maido observations to quantitatively evaluate the new and old turbulent schemes in FLEXPART will be addressed by the results of the OCTAVE project and will merit a separate publication rather than serve as a validation of the current presented work. Although it is correct that the

well-mixedness criterion on its own does not select the 'best' setup, it does confirm the plausible use of different configurations. It is however clear that the traditional turbulent time step configuration is incompatible with the TKE field ingestion. The choice of $L\_w$ parametrisation depends on which one is used in the NWP, the whole point is to get consistent results between the Eulerian and the offline model so it does not make sense to choose one offline, independent of the NWP. Concerning the Step TKE or SDA configurations, users are free to choose which one suits their needs best. One is not 'better' as the other. In general, when users are interested in a vertical output from FLEXPART-AROME that has a resolution similar or larger than the NWP we suggest using the SDA configuration with elevated values of CTL and IFINE. If the outgrid vertical resolution is lower than that of the NWP we suggest using the Step TKE configuration with slightly lower values of CTL and IFINE to speed up the simulation. The values of these input parameters are essential to the validity of the small discontinuity approximation.

Structure and Presentation: It is not always easy to follow the flow of the manuscript. It is often not well explained why and how certain things were done (see examples below). In other sections the manuscript is lacking the degree of detail that is important for a model development paper. Also, the current conclusions are lacking a clear recommendation, which of the new turbulence schemes should be used in future studies, and if the developments presented here will make it back into the main and/or WRF FLEXPART versions.

- We have added details and cleared up things as much as possible. There is not a single turbulent setting which can be systematically recommended (a fact that is also true for parameterizations available in mesoscale models) but rather four possible options from which the user should choose depending on the configuration of the NWP, the desired vertical resolution of the output and of the CTL and IFINE parameters as discussed above.

Minor comments: Abstract, last sentence: This was said before. Maybe reformulate to

make it the main conclusion of the study. Also include a statement/recommendation of the default turbulence scheme to be used with FLEXPART-AROME.

- Thank you for the suggestion, we changed the last part of the abstract.

Introduction: The pros and cons of Lagrangian versus Eulerian models are stated. However, this section is lacking good citations and it is also a bit too negative about Eulerian models. For example advection schemes in Eulerian models can be designed in ways that they are conserving mass and are less diffusive. But usually this comes with a prize of higher complexity and larger computational costs. But generally the statement that Eulerian models cannot do this is not valid. Also, there are other uncertainties connected to offline Lagrangian transport models. 1) temporal resolution of input meteorology, when running off-line, 2) less explicit description of turbulence (in comparison to prognostic TKE in most NWPs), exactly what the manuscript highlights later.

- We have reduced the imbalance between Eulerian and Lagrangian models, added a reference and reorganised the manuscript to highlight the difference in turbulent parameterizations in the introduction.

P2,L6ff: Regional inverse modelling studies are also an increasingly important field of application of LPDMs: eg. Stohl et al. (2009), Lin et al. (2003), Manning et al. (2003)

- Thank you for this note, it has been incorporated in the manuscript.

P2,L11f: Does this sentence still refer to different FLEXPART versions? Please mention these again with reference. Next to FLEXPART-WRF, there is also the FLEXPART-COSMO version mentioned in Pisso et al. (2019) and described in Henne et al. (2016).

- The manuscript has been adjusted.

P2,L15: "French metropolitan area" Does this refer to Paris or the whole mainland France domain?

Interactive
comment

- The French metropolitan area refers to the whole mainland of France. It is clarified in the new version.

P2,L20: Is AROME-SWIO also an operational model product by MeteoFrance?

- AROME is operationally used over the South-West Indian Ocean by Meteo-France. It is referred to in as AROME-IO or AROME-SWIO. Labelling issues aside, it is an operational model product of Météo France. I've adjusted the manuscript to make clear that it concerns an AROME configuration in the SWIO area to avoid the explicit labelling.

P2,L24: Reference to FLEXPART-WRF publication missing. On which FLEXPARTWRF version is FLEXPART-AROME based?

- FLEXPART-AROME is based on FLEXPART-WRF version 3.1.3.

P3,L9f: The sentence is incomplete. I guess it should continue after (fig 1) without starting a new sentence.

- Indeed, thanks for the remark.

PBL diagnostics: Why is the PBL height diagnostic, which was solely based on theta_v, called robust as compared to the Richardson bulk number approach in FLEXPART? The latter is also using the theta_v profiles from AROME, but in addition it also uses wind shear information (again from AROME). Compare Stohl et al. (2005). However, FLEXPART in its original version uses an "enveloping" PBL height, which is the maximum from the neighboring grid cells and the two model time steps in memory. It also extends the PBL height in areas with large subgrid-scale orographic variability. Both approaches may NOT be justifiable for high resolution simulations and may be the reason for the "overestimation" of PBL heights by FLEXPART in mountainous terrain (P3,L11). Which approach was followed in FLEXPART-AROME? The same as in FLEXPART ECMWF? It is also not clear how PBL heights were estimated solely based on theta_v. By a parcel method? Assuming that the PBL height is the height where theta_v is first

larger than theta_v at the surface including a surplus surface temperature? Was an additional interpolation between model levels used (like in FLEXPART)? - The virtual potential temperature method is called robust since it is a simple and straightforward diagnostic, not to contrast with the parametrisation in FLEXPART. The Richardson bulk number approach in FLEXPART is one of the many possible PBL height parametrisations. We do not want to say that one is better or worse than the other, rather that there is a problem in the number of ways that can be used to determine the PBL height. As an illustration we used a simple, straightforward and robust diagnostic to estimate the PBL height to highlight the fact that PBL tops can differ between parametrisations. If the offline diagnostic is in disagreement with the NWP turbulence it will misrepresent transport of air masses near the top of the boundary layer as is stated in the paper. By using the TKE from the model we no longer depend on offline parametrisations and improve consistency between the models which is the main goal of the development presented here. The PBL height from FLEXPART as shown in the manuscript does use the 'enveloping' scheme but the subgrid-scale orographic impact is switched off by putting the LSUBGRID input parameter to zero as it indeed is not justified in AROME grid resolution. - The PBL height based on theta_v is determined at the level where the virtual potential temperature equals the one at the surface with a certain surplus. There is no interpolation between model levels used. We have adapted the discussion to a comparison between the FLEXPART boundary layer height ant the TKE fields from AROME for simplicity's sake.

Comparing turbulent layers and PBL heights: The TKE layer diagnosed from AROME is strictly speaking not the same as the classical PBL height. Hence, I suggest to clearly separate the naming from what is otherwise called PBL height. This is implicitly introduced in the description, but it would be better to clearly distinguish between this turbulent layer and the PBL! It would be good to clearly define this layer in section 2 and explain in more detail how it was diagnosed from the model output. Currently this is only done in the caption to Figure 2 although the resulting layer depth is already displayed in Figure 1. From this it is also clear that one cannot conclude from the comparison

between turbulent layer and PHL heights that the latter is under- or overestimated (last sentence section 2). One can only say that the former is greater or smaller than the other.

- To avoid confusion, we changed the discussion where we only compare the PBL height from the FLEXPART parametrisation with the TKE fields, as mentioned above.

P3,L15: In FLEXPART it is also possible for particles to cross from the PBL to the FT through the subgrid-scale convection scheme. Was this switched on or off for FLEXPART-AROME. The scale would probably call for switching it off but this was not clearly stated anywhere?

- The FLEXPART subgrid scale convection concerns deep convective motions. This is resolved at the AROME resolutions and is turned off by setting the LCONVECTION input parameter to zero.

P3,L21: This "erratic behavior" could be avoided by detecting the layer top where at least two neighboring levels show TKE below the threshold value. From the examples given in Fig 2. It seems it is always a single level with low TKE between PBL and shallow convection zone. Does the erratic behavior actually matter for the TKE-based turbulence scheme in FLEXPART-AROME? Or is it only a diagnostic for the comparison with FLEXPART's PBL height estimation? This should be clarified in the text as well.

- The height of the turbulent layer in AROME does not influence the transport in FLEXPART-AROME but is rather a diagnostic from AROME output most comparable to the PBL height, which is why it was shown here. In FLEXPART-AROME the turbulent layer height is not used in any way.

One last question concerning the TKE layer height: From the layer heights displayed in Fig. 1, my conclusion would be that mixing over the sea is more intensive or at least reaches higher than over the mountains. This is counter-intuitive, but possibly related to the fact that heights above ground are shown. What are the model orography heights

for the 4 points for which the layer heights were evaluated in Fig 1?

- The surface level in the mountains is 1222 m above sea level. The counter-intuitive conclusion that mixing above sea reaches higher altitudes is indeed due to the fact that heights are shown in meter above ground level.

P5,L15: At this point not clear what a Thomson interface is and there is no reference given either.

- This is indeed a specific term that is not referenced anywhere else in the paper. It is a name we use for the interfaces at which the particles can reflect and is described elsewhere in the manuscript as 'TKE interfaces'. We have changed this here.

P5,L15 and section 3.1: How does the Thomson approach actually justify setting the density correction to zero? The density is not affected by turbulence intensity and as such density gradients are not explicitly treated by the Thomson approach.

- The density correction is not set to zero, only the drift correction. Thank you for the remark. As said in section 3.1, Lin et al. (2003) did include a density correction in their implementation of the method proposed by Thomson et al. (1997) but we opted to keep the FLEXPART density correction. Both possibilities were tested and had identical results. We decided to keep the FLEXPART density correction implementation.

P5,L18ff: Point out that this choice refers to the 1D, 3D options in Table 1. Please give a reference to the "diagnostic equations from Meso-NH" so that these can be found by the interested reader or even repeat them here if they are central.

- Turbulent motions implemented in Meso-NH can be found in Cuxart et al. (2000). They are not central since they are not specifically tested in this work. We have added the reference. Thank you for the remark.

P7,L3ff: The two different ways how to calculate the time step should be introduced much clearer and the terms 'bottom-up' and 'top-down' properly introduced as two ways how to calculate the turbulent time step. These terms have different meanings in

different fields and in the context of the time step it remains a bit unclear why they were chosen.

- We have rewritten this part and changed the terminology to adaptive vertical turbulence time step (FVTTS) versus a fixed vertical turbulence time step (AVTTS).

Turbulent mixing length: Would it be possible to give the equations for the three different ways how L_w was calculated?

- We have added the implementation of these parametrisations in the annexes of the manuscript.

Section 4, Validation: The setup should be described with more care and detail. For example: Were mean wind fields set to zero for this case? Were actual fields from AROME used for this exercise or some standard fields? Does it matter which exact locations were chosen? The location should not matter, only the surface properties, if winds were set to zero. How many particles were used in these exercises? How many per grid column? What was the FLEXPART PBL height in these grid cells and how did the TKE profile look like. Maybe both could be added to Fig 4.

- Vertical turbulent motions were isolated by taking out displacements along the resolved winds. The TKE fields used for the tests are 3D fields obtained from AROME. The tests were run in a single column with 250000 particles. Different locations give similar results but since the TKE profiles differ between grid cells we specified which grid cell was chosen for the tests. A figure showing the TKE profiles and the PBL top evolution was added to the manuscript.

Section 4: The 3D configurations of the turbulence scheme are never discussed only introduced in section 3. Were these schemes not tested after all? If not, why introduce them in the methods sections? If they were tested, how did the results compare to the 1D cases?

- The 3D turbulent modes were not validated as the 1D modes as we have no AROME

runs with 3D turbulence. These modes were implemented to anticipate future developments and were only checked to not provide problems when running the software.

P8,L26: 'TURB_OPTION=11 and 111'. In the previous sentence the settings were spelled out not just the option index given. To keep the text flowing the same should be done here. The option index could still be given in braces (wherever this helps to clarify things). The sentence is also a bit odd, because already the previous sentence said 'DEARDORFF [...] has the least accumulation', which is the same as 'best conserve well-mixed state'. Better start with 'Besides the DEARDORFF modes, modes xxxx best coserve ...'.

- Thank you for the remark, we have adjusted the manuscript.

P8,L29ff: The discussion on L_w could be more illustrative if profiles of L_w could also be included in Fig 4 or together with PBL heights and TKE profiles in a supplement.

- We agree that showing values of L_w would be more illustrative. Unfortunately, since this is a local value that is recalculated at every step for each particle we cannot retrieve this kind of profile.

Figure 4/5: Why is the second WRF TKE mode called 'stable repartitioning' here? It was introduced as a 'TKE only' method above. No mention of it being stable. In the caption it should be repeated that these are results for a grid cell over the ocean.

- The second WRF TKE method is indeed based only on the TKE value without the PBL parameterisation used in FLEXPART. The stable repartitioning refers to the parametrisation used by FLEXPART-WRF to distribute the TKE over 3 dimensions.

P9,L4: I think this should be 'Near-surface concentrations".

- Indeed, thank you.

Section 4, Fig 5: How is it possible that the vertical gradients in the simulations with new turbulence scheme are maintained over time. Even if the mixing in the shallow

convection zone is smaller than in the PBL, one would expect that the vertical gradient eventually vanishes in the 24 hours of simulation, as it quickly does in the Hanna case within the PBL. In order to illustrate this more clearly, it would also be interesting to see the final mixing ratio profiles of all configurations in a comparison.

- The gradients are maintained over time due to the use of dynamic TKE profiles obtained from AROME. At certain times mixing reaches higher altitudes after which these particles are not mixed further. We have added final mixing ratio profiles in an annex as it is indeed interesting to see but does not contain new information compared to the plots shown in Figure 5.

Section 5: It would be valuable if this section would show some kind of comparison with observations at Maido. Such a comparison could help supporting the authors suggestion that the new turbulence modes are superior to previous schemes. Without such a comparison there is little evidence that the performance of the new schemes is more realistic.

- As stated before, the idea of these new developments were to get consistency between the offline transport model and the NWP. We don't claim that one is superior to the other but rather that we should not have a difference between the online and offline turbulent parametrisations for the reasons mentioned above.

P10,L3: It is always confusing with LPDMs to write about 'particle transport' which can easily be confused with aerosol particle transport. Here, it would probably work best to simply replace 'particle transport' by 'atmospheric transport'.

- Thank you for the suggestion, we have changed the wording.

Figures 6/7: Most of the text of the figure captions is also stated in the main text. Remove from caption. Rather repeat what is seen in each sub-panels. Sub-panels should also be labeled by letters.

- The manuscript has been adapted.

Section 5.2: This section needs some introduction on how computation times were estimated. Were repeated runs carried out for each configuration? This is important since run-times may differ due to other processes running on the same machine and/or I/O may be influenced by other processes. It would also be helpful to mention on what architecture and with which compiler (options) and with which parallelisation approach these results were obtained. Do these timings reflect run times for the complete model runs or just for the transport part of the model? Please speculate why the increase in computation time was so much larger for the well-mixed test compared to the surface release? Why were computation times larger for the no-turbulence cases? Shouldn't these perform much faster, since only the mean motion needs to be solved for (which was zero in the well-mixed and point release tests)? I see it is explained later on. So only a quick-and-dirty implementation of no turbulence was used. But then one should not compare these run-times. There is little value in it since they present some kind of artificial, never-used option.

- No repeated runs were carried out for each simulation as the simulations were run on a table-top machine which was dedicated to the FLEXPART-AROME simulations during the test phase. Simulations were run on a workstation with a single CPU INTEL CORE I7-7700, 32 Gb of DDR4 SDRAM with a GNU compiler. The machine was dedicated to the FLEXPART-AROME simulations to minimise the impact of parallel processes on the computation times. Currently the code is not parallellised but it is foreseen to update this and to use the openMP approach similar to the FLEXPART-WRF code. Run times reflect the complete model runs. We agree that the implementation of no-turbulence is quick-and-dirty. However, since it is the way it is implemented in the FLEXPART-WRF code and it is a possible choice of input by the user, we decided to include it in the comparison.

Conclusions: These are a bit non-conclusive. So what is the recommendation for future use of the model? Which turbulence mode should be used and why?

- We have adjusted the manuscript to reflect the considerations that were addressed in

previous answers.

P13,L6: Shouldn't this be 3D TKE fields? Which dimension would be dropped out for them to be 2D?

- Indeed, this is a typo. Thank you.

Technical comments:

- Thank you for the comments, it has been fixed.

[revised manuscript text omitted]